# Effects of Different Dietary Protein Level on Growth Performance, Rumen Fermentation Characteristics and Plasma Metabolomics Profile of Growing Yak in the Cold Season

**DOI:** 10.3390/ani13030367

**Published:** 2023-01-21

**Authors:** Yanbin Zhu, Guangming Sun, Luosang Dunzhu, Xin Li, Luosang Zhaxi, Suolang Zhaxi, Cidan Yangji, Basang Wangdui, Feng Pan, Quanhui Peng

**Affiliations:** 1Institute of Animal Husbandry and Veterinary Medicine, Tibet Academy of Agriculture and Animal Husbandry Science, Lhasa 850009, China; zhuyanbin126@126.com (Y.Z.); sungm1992@163.com (G.S.); sontsa76@163.com (L.D.); 13889092363@163.com (C.Y.); 2Linzhou Animal Husbandry and Veterinary Station, Lhasa 850009, China; leexin0828@126.com (X.L.); zhaxils@163.com (L.Z.); luozha202211@163.com (S.Z.); xuelangsuolang@163.com (S.); ciyang0414@163.com (C.); 3Institute of Animal Nutrition, Key Laboratory of Bovine Low-Carbon Farming and Safety Production, Sichuan Agricultural University, Chengdu 611130, China; 2020314070@stu.sicau.edu.cn

**Keywords:** growing yak, plasma metabolomics, rumen fermentation, dietary protein, cold season

## Abstract

**Simple Summary:**

The yak (*Bos grunniens*) is endemic to the Qinghai-Tibetan Plateau, and data on their nutritional requirements are lacking. Previous reports have shown that yaks have a higher nitrogen utilization efficiency than *Bos taurus*, so we hypothesize that yak diets should have a lower level of dietary crude protein than beef cattle. Two diets with different dietary protein content were designed to investigate the effect of dietary protein levels on yaks in the cold season. The results showed that during the cold season, high protein level could promote rumen microbial protein synthesis, elevate arginine and proline metabolism, purine and pyrimidine metabolism, taste transduction and glutathione metabolism pathways, enhance antioxidant and immune function, and promote growth performance of yak. However, a high-protein diet enhanced the renin secretion pathway, which might increase a yak’s water intake.

**Abstract:**

This experiment was aimed to compare the effects of two diets with different protein content on the growth performance, immune indexes, rumen fermentation characteristics and plasma metabolomics of growing yak in the cold season. A total of 24, 2-year-old healthy yaks with similar body weight (142.9 ± 3.56 kg) were randomly allocated to two isoenergetic diets with different protein content (10 vs 14%) according to a non-paired experimental design, and the protein of the diets was increased by increasing soybean meal, rapeseed meal and cottonseed meal. The growth performance experiment lasted 56 days. Four days before the end of the growth experiment, the digestion trial was conducted, and the rumen fluid and plasma was collected for measurement. The results showed that the average daily feed intake (*p <* 0.001) and average daily gain (*p* = 0.006) of yak fed a high-protein diet was significantly greater, while the feed conversion ratio was lower (*p =* 0.021) than that of yaks fed a low-protein diet. Plasma aspartate aminotransferase (*p =* 0.002), alanine aminotransferase (*p <* 0.001), malondialdehyde (*p =* 0.001), tumor necrosis factor-α (*p* = 0.032) and interferon-γ (*p* = 0.017) of the high-protein group were significantly lesser, whereas superoxide dismutase (*p* = 0.004) and interleukin-2 (*p* = 0.007) was significantly greater than that of the low-protein group. The rumen microbial crude protein (*p* < 0.047) and crude protein digestibility (*p* = 0.015) of yak fed a high-protein diet was significantly greater than that of the low-protein group. The metabolomics results showed that yaks fed a high-protein diet were elevated in protein digestion and absorption, arginine and proline metabolism, tryptophan metabolism, purine metabolism, butanoate metabolism, taste transduction, pyrimidine metabolism, pantothenate and CoA biosynthesis, glutathione metabolism and renin secretion pathways. It is concluded that a high-protein diet in the cold season can promote rumen microbial crude protein synthesis, enhance antioxidant and immune function and promote growth performance of yaks.

## 1. Introduction

The yak is a species endemic to the Qinghai-Tibetan Plateau, and is an important means of production and subsistence for the herders on the plateau and can provide herders with necessary materials, such as meat, milk, wool, skins and fuel [1]. China’s yak stock is over 15 million, accounting for 95% of the world’s yaks, mainly distributed in Tibet, Qinghai and Sichuan province [2]. With the development of social economy, yak farming has gradually changed from the all-grazing mode to the three-dimensional farming mode, which comprises all-grazing in high-altitude zones, grazing and supplementing feeds in mid-altitude zones, and fattening indoor in low-altitude zones.

Because the proportion of indoor fattening yak is increasing year by year, there has been research on the effects of different nutritional levels on indoor-fed yak [3,4]. Possibly, in plateau areas, especially in the cold season, energy intake is more important than protein in terms of sustaining life; therefore, the research is mainly focused on the effects of the energy level in yaks [3,5], whereas the effects of protein in yaks are scarce [6,7]. It has been reported that the reason why yaks can adapt to high altitudes and extreme cold areas and survive well with a shortage of forage in the cold season is because yaks can use dietary protein more efficiently than other species [5,6], and therefore, their protein requirements may be lower than other species. However, the several existing studies on the effects of dietary nitrogen levels on yak are mainly mechanistic studies using digestion and metabolism trials [6,7], whereas the effect of protein levels on the growth performance of growing yaks in the cold season has not been reported.

The metabolome is in the most downstream of systems biology, and the subtle functional changes of the genome and proteome can be amplified at the metabolic level, and the detection is easier. The number of metabolites of animals is much less than that of genes and proteins, and the common metabolites are very similar in different biological systems, so metabolomics can be applied to different biological systems. Taking advantage of this, Zhao et al. [8] compared the rumen microbial metabolic pathways difference among Qaidam yellow cattle, dzomo and yak, and found that amino acids, carboxylic acids, sugars and bile acids were different, and these changes were attributed to the rumen microbial composition. Xue et al. [9] compared the yak serum metabolomics profile difference after supplementing hull-less barley versus rapeseed meal in the warm season, and reported that supplementation of yak with barley was more efficient in the promotion of yak glucose and protein anabolism compared to supplementation with rapeseed meal. However, until now, the metabolic response of indoor-fed growing yak to different dietary protein levels in the cold season is unclear.

Therefore, two diets with different protein levels were formulated in this experiment, with the aim to investigate the effects of dietary protein content on the growth performance, plasma immunity and antioxidant indexes, rumen fermentation characteristics and plasma metabolic profiles in growing yaks in the cold season.

## 2. Materials and Methods

### 2.1. Ethic

The experimental protocol used in the present study was approved by the Animal Policy and Welfare Committee of the Agricultural Research Organization of Tibet Autonomous Region, China, and was in accordance with the guidelines of the Animal Care and Ethical Committee of the Institute of Animal Husbandry and Veterinary Medicine, Tibet Academy of Agriculture and Animal Husbandry Science (TAAAHS-2020–172).

### 2.2. Study Location

The experiment was conducted at the Linzhou yak breeding demonstration farm, Linzhou County, Lasha City, Tibet (29°45′ to 30°08′ N, and 90°51′ to 91°28′ E) at an altitude of 3353 m. This study was carried out during October and December 2021. The average temperature was recorded as 0.6 °C, ranging from –18 °C to 25 °C. The average humidity was approximately 40% during the experimental period.

### 2.3. Animals, Experimental Design and Diets

In total, 24, 2-year-old yaks (*Bos grunniens*), half male and half female, were allocated to one of two dietary treatments according to a non-paired design and according to body weight. The animals were either fed a diet with low protein (10%, LP) or high protein (14%, HP), total mixed ration (TMR) *ad libitum*. The ingredients and chemical composition of the diets is shown in Table 1. Prior to the beginning of the formal trial, all animals were assigned to an individual pen (7 m × 5 m) and had a 15-day adaptation period. The formal growth trial lasted 56 days. The TMR was provided twice a day at 09:00 and 17:00 h at equal quantity, and animals had free access to fresh water during the whole experimental period.

### 2.4. Growth Performance

At the beginning of the formal experiment, all the animals were weighed and initial body weight (IBW) was recorded, and on the morning of final day, all the animals were individually weighed again, after which the final body weight (FBW) was recorded. The average daily gain (ADG) was calculated accordingly. The amount of feed provided and leftover per yak per day was recorded and the average daily feed intake (ADFI) was calculated based on the dry matter content. Finally, the feed conversion ratio (FCR) was calculated on the basis of ADFI and ADG.

### 2.5. Nutrients’ Apparent Digestibility

In order to reduce pollution, before we carried out the digestion trial, a thick plastic film was spread on the floor of the pen. During the 52–56 d, the total feces were collected into the plastic bucket with a small shovel immediately after yak defecation. The feeds and approximately 5% of the total feces were collected before the 09:00 h feed. The daily collected feces were retained and combined and 10% sulfuric acid was added to prevent nitrogen loss. The dealt feces samples were stored at −20 °C for later analysis.

Feed and fecal samples were analyzed for dry matter (DM) (method 930.15), acid detergent insoluble ash (method 935.29) and crude protein (CP) (method 984.13) according to the methods of the Association of Official Analytical Chemists [11]. Neutral detergent fiber (NDF) and acid detergent fiber (ADF) were analyzed according to the Van Soest et al. [12] method using heat-stable α-amylase and expressed without residual ash.

### 2.6. Plasma Parameters

On the morning of day 56 before feeding, approximately 10 mL blood was sampled from the jugular vein. The anticoagulant in the vacutainer tubes was ethylene diamine tetraacetic acid. Then, the blood samples were centrifuged for 15 min at 4 °C at 3000× *g*, and plasma was carefully stored at −20 °C until further use. Plasma samples were analyzed for total protein (TP), albumin (ALB), globulin (GLOB), glucose (GLU), triglyceride (TG), blood urea nitrogen (BUN), alanine aminotransferase (ALT), aspartate aminotransferase (AST) and alkaline phosphatase. (ALP) using an automatic biochemical analyzer (Hitachi 7200, Hitachi Group, Tokyo, Japan).

The concentration of glutathione peroxidase (GSH-Px), superoxide dismutase (SOD), total antioxidant capacity (T-AOC) and malondialdehyde (MDA) were determined using the standard commercial kits (Jiancheng Bioengineering Institute of Nanjing, Nanjing, China) according to the manufacturer’s instructions.

The measurement of plasma immune indexes interleukin 2 (IL-2), IL-6, IL-10, interferon-γ (IFN-γ), tumor necrosis factor α (TNF-α), Immunoglobulin A (IgA), IgM and IgG were carried out using ELISA kits (Sigma Chemical Co. Shanghai, China). Ophenylenediamine was used as the substrate and absorbance was measured at 450 nm with an ELISA reader (Bio-Tek Instruments Inc., Winooski, VT, USA).

### 2.7. Plasma Metabolomics Profile

The Agilent 7890 gas chromatograph system coupled with a Pegasus HT time-of-flight mass spectrometer (LECO, St, Joseph, MI, USA) was used to conduct GC/MS analyses of plasma samples. In total, 100 μL of each sample was firstly mixed with 370 µL of solvents composed of 350 µL methanol and 20 µL l-2-chlorophenylalanine (0.1 mg/mL stocked in dH_2_ O), after which the mixture was vortexed for 10 s and centrifuged for 15 min at 3000× *g*, at 4 °C. The supernatant (0.34 mL) was transferred into a fresh GC–MS glass vial, and 12 µL supernatant of each sample was taken and pooled as a quality control (QC) sample. All samples were firstly dried in a vacuum concentrator without heating and then incubated for 20 min at 80 °C after adding 55 µL of methoxy amination reagent (20 mg/ mL dissolved in pyridine) into each sample. In total, 75 μL of BSTFA reagent (1% TMCS, v/v) was added to each sample then all samples were incubated for 1 h at 70–80 °C. Subsequently, 10 µL Fatty Acid Methyl Ester (FAMEs) (standard mixture of fatty acid methyl esters, C8–C16:1 mg/mL, C18–C24:0.5 mg/mL in chloroform) was added to each sample after all samples were cooled to room temperature. After adding all reagents, each sample was mixed well for GC–MS analysis. In total, 1 μL of the analyte was injected into a DB5 MS capillary column coated with 5% diphenyl cross-linked 95% dimethylpolysiloxane (30 m × 250 µm inner diameter, 0.25 µm film thickness, J&W Scientific, Folsom, CA, USA). In total, 1 μL of the analyte was injected in splitless mode. Helium was used as the carrier gas. The front inlet purge flow was 3 mL/min, and the gas flow rate through the column was 20 mL/min. The initial temperature was kept at 50 °C for 1 min, then raised to 320 °C at a rate of 10 °C/min. The temperature was kept for 5 min at 320 °C. The injection, transfer line and ion source temperatures were 280, 280 and 220 °C, respectively. The energy was −70 eV in electron impact mode. The mass spectrometry data were acquired in full-scan mode with the m/z range of 85–600 at a rate of 20 spectra per second after a solvent delay of 366 s.

### 2.8. Rumen Fermentation Characteristics

On day 56, a stomach tube was used to collect approximately 200 mL of ruminal fluid 2, 4 and 6 h post morning feeding. Four layers gauze was used to separate the ruminal fluid from the feed particles, and the pH of ruminal fluid was measured immediately using pH meter (PHS-3 C, Shanghai, China). Thereafter, the rumen fluid was centrifuged for 15 min at 1200× *g*. The perchlorate liquid was used for the removal of protein, and liquid samples were then stored at −20 °C until further use. Potassium hydroxide was used to neutralize the fluid and centrifuged for 10 min at 400× *g* for the determination of ammonia nitrogen (NH_3_-*n*) and microbial protein (MCP), and for the determination of volatile fatty acid (VFA) analyses, another perchlorate liquid sample was used. VFA analysis was performed using an HPLC organic acid analysis system (Shimadzu, Kyoto, Japan) The supernatant was shaken with cation exchange resin (Amberlite, IR 120 B H AG, Organo Corporation, Tokyo, Japan) and centrifuged at 6500× *g* for 5 min. The supernatant was passed through a 0.45 μm filter under pressure, and the filtrate was then injected into an HPLC system. The analytical conditions were as follows: column, SCR-101 H (7.9 mm × 30 cm) attached to a guard column SCR (H) (4.0 mm × 5 cm) (Shimadzu, Tokyo, Japan), oven temperature, 40 °C, mobile phase, 4 mM *p*-toluenesulfonic acid aqueous solution, reaction phase, 16 mM Bis-Tris aqueous solution containing 4 mM *p*-toluenesulfonic acid and 100 μM ethylenediaminetetra-acetic acid, flow rate of the mobile and reaction phase, 0.8 mL/min, detector and conductivity detector (CDD-6 A, Shimadzu). NH_3_-*n* was measured as described previously by Chaney et al. [13], and MCP was determined according to the procedure described by Makkar et al. [14].

### 2.9. Statistical Analyses

The results of the growth performance, plasma indexes, rumen fermentation characteristics and nutrients digestibility were analyzed using *t*-student test and presented as the least-squares means with the standard deviation of the means. Significance was described as *p* ≤ 0.05, while 0.05 < *p* < 0.10 was regarded as a trend. For metabolomics data analysis, Chroma TOF 4.3 X software of LECO Corporation and LECO-Fiehn Rtx5 database were used for raw peaks exacting (Joseph, Michigan, USA), data baselines filtering and calibration of the baseline, peak alignment, deconvolution analysis, peak identification and integration of the peak area. Multivariate analysis including principal component analysis (PCA) and orthogonal correction partial least squares discriminant analysis (OPLS-DA) were conducted using SIMCA-*p* software (V 14.0, Umetrics, Umea, Sweden). Differentially expressed metabolites between two treatments were identified based on variable importance in projection (VIP) from OPLS-DA analysis and statistical analysis (VIP > 1 and *p* < 0.05 Kyoto Encyclopedia of Genes and Genomes (KEGG, http://www.genome.jp/kegg/ (accessed on 11 January 2023)) was conducted to view the enriched pathways of different metabolites.

## 3. Results

### 3.1. Effects of the Dietary Protein Level on the Growth Performance of Growing Yaks in the Cold Season

Table 2 shows the effects of the dietary protein level on the growth performance of growing yaks in the cold season. The FBW (*p* = 0.018), ADG (*p* = 0.006) and ADFI (*p* < 0.001) of the high-protein group was greater than the low-protein group. On the contrary, the FCR was lesser (*p* = 0.021) than the low-protein group.

### 3.2. Effects of the Dietary Protein Level on the Nutrients’ Apparent Digestibility of Growing Yaks in the Cold Season

Table 3 shows the effects of the dietary protein level on the nutrients’ apparent digestibility of growing yaks in the cold season. The digestibility of the CP (*p* = 0.015) and NDF (*p* = 0.047) of the high-protein group was greater than that of the low-protein group, but no significant difference was observed in the apparent digestibility of DM, OM and ADF (*p* > 0.05).

### 3.3. Effects of the Dietary Protein Level on the Plasma Biochemical indexes of Growing Yaks in the Cold Season

The effects of the dietary protein level on the plasma biochemical indexes of growing yaks in the cold season are presented in Table 4. Compared with the low-protein group, the ALT (*p* < 0.001) and AST (*p* = 0.002) concentration of the high-protein group was lesser, whereas the UN (*p* = 0.038) and GLU (*p* = 0.004) was greater than that of the low-protein group. There was no significant difference in the TP, GLOB, TG and ALP (*p* > 0.05).

### 3.4. Effects of the Dietary Protein Level on the Plasma Antioxidant Indexes of Growing Yaks in the Cold Season

The effects of the dietary protein level on the plasma antioxidant indexes of growing yaks in the cold season are shown in Table 5. The SOD concentration of the high-protein group was greater (*p* = 0.004), whereas the MDA (*p* = 0.001) was lesser than that of the low-protein group. No significant difference was observed in T-AOC and GSH-Px (*p* > 0.05).

### 3.5. Effects of the Dietary Protein Level on the Plasma Cytokine and Immunoglobulin Content of Growing Yaks in the Cold Season

Table 6 shows the effects of the dietary protein level on the plasma cytokine and immunoglobulin content of growing yaks in the cold season. The IL-2 (*p* = 0.007) and IgG (*p* = 0.012) content of the high-protein group was greater, whereas the IFN-γ (*p* = 0.017) and TNF-α (*p* = 0.032) content was lesser than that of the low-protein group. No significant difference between the two groups was observed in the IL-4, IL-6, IL-10, IgA and IgM content (*p* > 0.05).

### 3.6. Effects of the Dietary Protein Level on the Rumen Fermentation Characteristics of Growing Yaks in the Cold Season

The effects of the dietary protein level on the rumen fermentation characteristics of growing yaks in the cold season are presented in Table 7. The MCP (*p* = 0.047) and TVFA (*p* = 0.008) content of the high-protein group was greater, whereas the NH_3_-*n* (*p* < 0.001) content was lesser than that of the low-protein group. There was a tendency for the rumen pH (*p* = 0.084) and butyrate (*p* = 0.088) of the high-protein group to be greater than that of the low-protein group.

### 3.7. Multivariate and KEGG Analysis

The PCA and OPLS-DA results are presented in Figure 1. All samples were scattered in the 95% confidence zone. Almost all of the samples from the two groups could be differentiated clearly, with the exception of XS6 5–7 (Figure 1A). The PC1 and PC2 account for 51% and 10.8% of the total variance, respectively. In the OPLS-DA, all the samples from the two different dietary protein groups could be differentiated clearly, and the PC1 and PC2 account for 32.2% and 23.5%, respectively, of the total variance (Figure 1B). The R2 X (the explanatory rate of the model in the X-axis), R2 Y (the explanatory rate of the model in the Y-axis) and Q2 (the predictive rate of the model) were 0.674, 0.998 and 0.844, respectively, which means that the model is valid and predictable (Figure 1C). Based on the differently expressed metabolites (Table 8), the KEGG analysis was performed (Table 9, Figure 2), and the results showed that the yaks fed diets with different protein levels showed differences in protein digestion and absorption, arginine and proline metabolism, tryptophan metabolism, purine metabolism, butanoate metabolism, taste transduction, pyrimidine metabolism, pantothenate and CoA biosynthesis, glutathione metabolism and renin secretion pathways.

## 4. Discussion

### 4.1. Growth Performance and Total Tract Digestibility

It has long been believed that yaks are more tolerant of extreme harsh environment than other animals, and their protein utilization efficiency is higher, so their protein requirements are lower [7,15]. However, the results of this experiment showed that increasing the protein level of the diet improved the ADG of growing yaks in the cold season. This phenomenon may be mainly due to that yaks that were fed with the high-protein diet increased the DM intake by 18%. This represents 18% more energy intake (56.40 vs 46.41 MJ/d). During the experiment period, the average temperature of the study location was 0.6 °C. It has been reported that the thermoneutral zone of the yak is 8–14 °C [16]; the low temperature during the experimental period increased the heat production, and hence, the maintenance requirement. The more daily energy is available, the less energy is used for maintenance. Therefore, lowered FCR was obtained in the high-protein group. In fact, previous studies have only compared the protein utilization efficiency of yak and yellow cattle [7,11], and did not focus on the effects of different protein levels on the growth performance of yaks. In addition, the results of our later metabolomics assays showed that the increased dietary protein content enhanced yaks’ plasma taste transduction pathway, which may be one the reasons for the improvement of ADFI. Increasing the dietary protein level improved the apparent digestibility of CP and NDF. On the one hand, diets had different ruminal degradable protein concentrations, and this affected the total crude protein digestibility. On the other hand, because metabolizable protein is diluted due to *n* endogenous contributions, the increase of the apparent digestibility between diets that contain different CP concentrations is more apparent [17]. In diets with lower CP, numerically, endogenous contributions become more important. Thus, increases in CP digestibility was mediated in part by the difference in RDP between treatments, and in part, by the difference in the crude protein level between treatments. This was in agreement with a previous report by Zhang et al. [7].

### 4.2. Plasma Parameters

In this experiment, it was observed that the blood glucose concentration of yaks fed a high-protein diet group was greater than the low-protein group, which meant that the yaks of this group metabolize more efficiently, because glucose is the most efficient energy supply. Yaks that were fed a high-protein diet consumed more feed and then 18% more energy. Increases in glucose were directly affected by energy intake. The BUN was significantly greater in the high-protein group than in the low-protein group, which may also be caused by a higher RDP, which was consistent with the results that were reported by Wu et al. [4]. Although elevated BUN implies a decreased renal function, the BUN concentrations observed in this study were all in the normal range. The lower ALT and AST concentration in the high-protein group meant improved liver function [4,18]. Furthermore, the pro-inflammatory factor TNF-α and IFN-γ concentration were reduced, and the IgG and anti-inflammatory factor IL-2 was elevated, which meant that the yaks in the high-protein group had a stronger humoral and cellular immunity function [19,20]. Furthermore, the SOD concentration of yaks fed a high-protein diet was greater, and the MDA was lesser than that of the low-protein group, which meant that the antioxidant capacity of the high-protein group was greater than the low-protein group [21]. The results of metabolomics also showed that the arginine and proline metabolism and the glutathione metabolism of the high-protein group were elevated, and the enhancement of these two metabolic pathways was expected to increase the antioxidant and immunity capacity of the yak.

### 4.3. Rumen Fermentation Characteristics

Greater MCP and NH_3_-*n* was observed in the high-protein group, in accordance with previous research [7,15]. A high protein, and hence, high soluble protein content, and low starch-to-protein ratio jointly led to an increase in the NH_3_-*n* concentration. The higher MCP was attributed to the greater quantity of the OM that is fermented in rumen. In this experiment, yaks fed a high-protein diet showed a greater OM intake, and theoretically, greater rumen OM digestibility, which could be supported by the greater quantity of ruminal TVFA produced, and a greater TVFA production has been reported in bulls fed a high-protein diet [22]. The results of the metabolomics assay showed that the purine and pyrimidine metabolism pathway of the high-protein group was enhanced, which suggested a strengthened rumen MCP synthesis was achieved.

### 4.4. Metabolic Pathway

The treatment effect of this study was the different dietary protein content; therefore, it was expected that the metabolic pathway of protein digestion and absorption was affected in the first place. The differently expressed amino acids that were identified in the plasma were valine, L-tyrosine, L-tryptophan, L-phenylalanine, L-methionine, L-isoleucine, L-glutamic acid and beta-alanine, which is consistent with the results reported by Kim et al. [23], who observed that the protein digestion and absorption metabolism was affected in Hanwoo cattle fed diets with different protein contents.

Subsequently, spermine, spermidine, *n*-carbamoylputrescine, L-glutamic acid, gamma-glutamyl-gamma-aminobutyraldehyde, gamma-aminobutyric acid, D-proline and creatinine were significantly changed. It has been reported that these metabolites are related to arginine metabolism [24]. In addition, D-proline, L-glutamic acid and gamma-aminobutyric acid were significantly changed, which meant that the proline metabolism was affected, and the strengthened arginine and proline metabolism pathway will promote the antioxidant and immune function of yak to some extent [25,26]. Beyond that, the xanthurenic acid, *n*-acetylserotonin, L-tryptophan, kynurenic acid, 1-benzazole, indole-3-ethanol, indole-3-acetic acid and 3-methyldioxyindole were also affected; all these are intermediate metabolites of tryptophan, which indicated that the tryptophan metabolism was affected, which was similar to previous research observed in dogs with increased dietary protein content [24].

The contents of xanthine, hypoxanthine, guanosine, guanine, adenosine, adenine, 2’-deoxyadenosine, (R) (-)-allantoin, uracil, thymine and cytidine were affected, which indicated that the purine and pyrimidine metabolism pathways were affected. This was also reported previously by Liang et al. [27], and Yin et al. [28] observed variations in purine and pyrimidine metabolism when bacteria were fed diets with different protein contents. The enhanced purine and pyrimidine metabolism of the high-protein group was attributed to the increased MCP synthesis in the rumen [29]. Zhou et al. [15] reported increased MCP in the rumen of yaks fed a high-protein diet, and Zhou et al. [15] and Tas and Susenbeth [29] reported increased urinary total purine derivatives, allantoin and uric acid with an increase in dietary nitrogen content [30].

The content of L-glutamic acid, γ-aminobutyric acid, acetone, (R)-Acetoin and (R)-3- ((R)-3-Hydroxybutanoyloxy) butanoate were affected, meaning that the butanoate metabolism pathway was changed, which was in agreement with previous reports [31]. Moreover, the increased butanoate metabolism implies that a high-protein diet might be helpful in promoting yak’s growth [32,33]. Because acetyl-CoA is the most important node substance of the TCA, the pantothenate and CoA biosynthesis pathway was enhanced in yaks fed a high-protein diet, which could suggest that more ATP was generated, and resulted in promoted growth performance [33,34]. The content of saccharin, norepinephrine, L-glutamic acid, gamma-aminobutyric acid and D-serine increased and the taste transduction pathway was enhanced, which might imply that the palatability of the high-protein diet was better than that of a low-protein diet under the conditions of this experiment; therefore, the ADFI of yaks fed a high-protein diet is greater compared to a low-protein diet. This phenomenon was proven previously by Rotzoll et al. [35]. When the protein content of the diet is not enough to meet the requirement, increasing the protein content can improve the palatability of the diet. The increased norepinephrine, epinephrine and adenosine contents implied that the renin secretion pathway was elevated, which will lead to thirst, and subsequently, increase the requirement of water. The greater need for water was also reported previously, in studies where animals were fed a high-protein diet [36,37].

## 5. Conclusions

Increasing the dietary protein enhances the plasma taste transduction pathway and increase the ADFI of yaks. Increasing the dietary protein promotes arginine and proline metabolism and the glutathione metabolism pathway, which resulted in improved antioxidant and immune capacity and growth performance. However, when yaks are fed a high-protein diet, the renin secretion pathway is enhanced, which might increase the animal’s water requirement.

## Figures and Tables

**Figure 1 animals-13-00367-f001:**
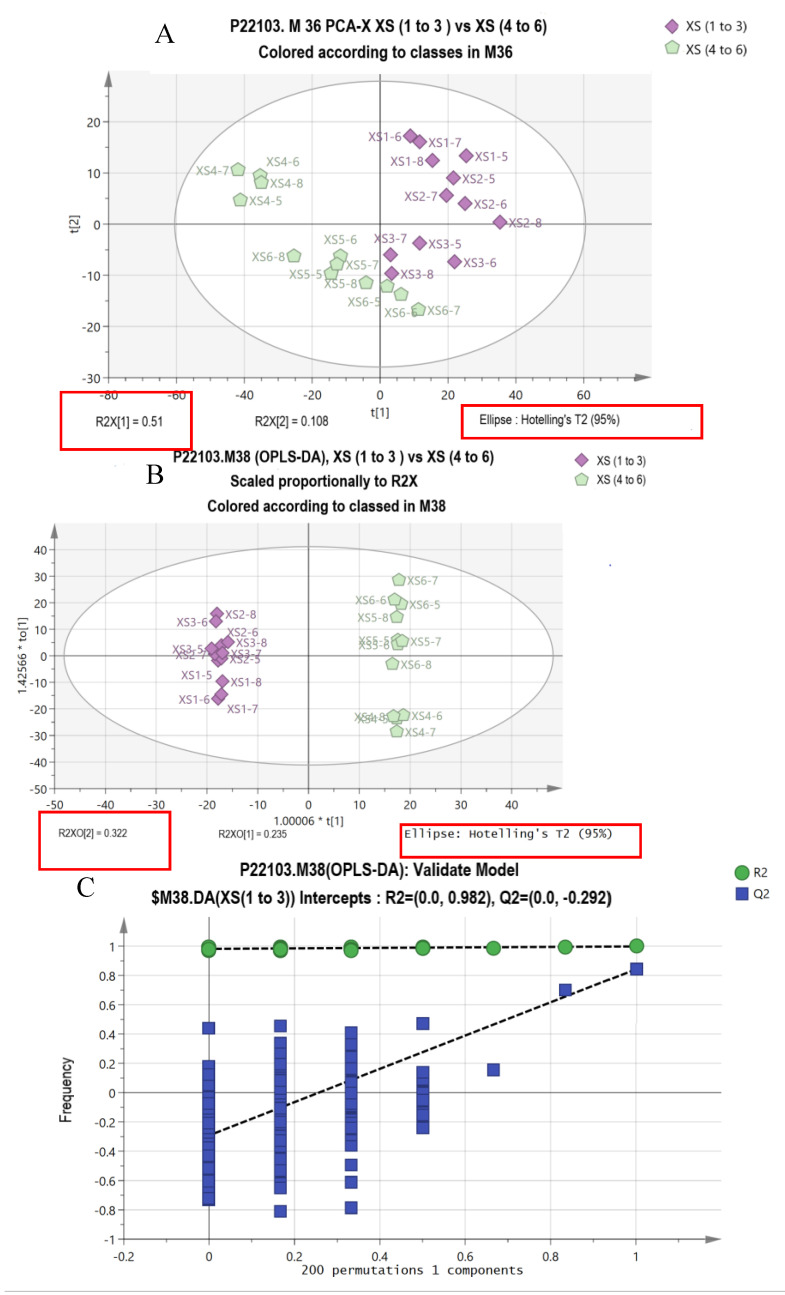
(**A**). Principal components analysis (PCA) of plasma metabolites from yak (*n* = 8) fed with a low- (1 to 3) and high-protein (4 to 6) diet. (**B**). Orthogonal correction partial least squares discriminant analysis (OPLS-DA) of the plasma metabolites from yak (*n* = 8) fed with a low- (1 to 3) and high-protein (4 to 6) diet. (**C**). OPLS-DA model permutation test chart.

**Figure 2 animals-13-00367-f002:**
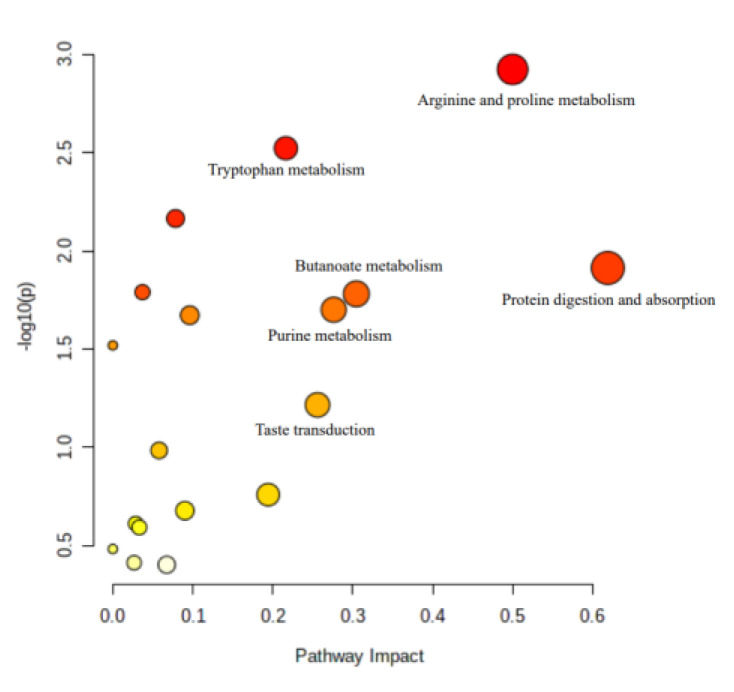
Metabolome view map of plasma metabolites from yaks (*n* = 8) fed a high-protein diet versus a low-protein diet. X-axis represents the pathway impact and Y-axis represents the *p* value. A larger size of the circle indicates more metabolites enriched in that pathway and a larger abscissa indicates higher pathway impact values. A darker color indicates smaller *p* values.

**Table 1 animals-13-00367-t001:** Ingredients and chemical composition of the experimental diets.

Item	Treatment	
	Low-Protein Group	High-Protein Group
Alfalfa hay	13.30	13.30
Oat hay	13.30	13.30
Full corn silage	13.40	13.40
Corn	48.00	43.00
Rapeseed oil	0.50	0.50
Wheat bran	4.00	3.00
Soybean meal	1.50	3.50
Cottonseed meal	1.50	3.50
Rapeseed meal	1.50	3.50
Calcium carbonate	1.44	1.43
Calcium hydrogen phosphate	0.56	0.57
Vitamin and minerals Premix ^1^	1.00	1.00
ME MJ/kg DM ^2^	8.58	8.55
CP %	9.91	13.89
RDP % CP	67.5	78.8
NDF %	26.84	27.21
ADF %	14.67	15.30
Ca %	0.83	0.81
*P* %	0.43	0.41

^1^ The vitamin and minerals premix provides per kg diet: 10 mg Cu in the form of sulfate, 60 mg Zn in the form of sulfate, 50 mg Mn in the form of sulfate, 50 mg Fe in the form of sulfate, 0.2 mg Co in the form of chloride, 0.5 mg I in the form of iodate, 0.3 mg Se in the form of selenite and, Vitamin A 10,000 IU, Vitamin D_3_ 2000 IU, and Vitamin E 50 IU. ^2^ ME value of TMR was calculated based on the available ME data of the ingredients used [9]. DM, CP, NDF, ADF, Ca and P were determined values [10]. DM, dry matter; ME, Metabolizable energy; CP, Crude protein; NDF, Neutral detergent fiber; ADF, Acid detergent fiber; Ca, Calcium; *p*, Phosphorus.

**Table 2 animals-13-00367-t002:** Effects of the dietary protein level on the growth performance of growing yaks.

Item	Treatment		*p*-Value
	Low-Protein Group	High-Protein Group	
IBW kg	142.3 ± 2.436	143.7 ± 3.044	0.727
FBW kg	167.2 ± 2.670	178.3 ± 3.407	0.018
ADG kg	0.44 ± 0.034	0.62 ± 0.045	0.006
ADFI kg/d	5.41 ± 0.140	6.60 ± 0.44	<0.001
FCR	12.30 ± 0.588	10.65 ± 0.694	0.021

IBW, initial body weight; FBW, final body weight; ADG, average daily gain; ADFI, average daily feed intake; FCR, feed conversion ratio. Values are presented as the least-squares means with the standard deviation of the means.

**Table 3 animals-13-00367-t003:** Effects of the dietary protein level on the nutrients’ apparent digestibility of growing yaks.

Item	Treatment		*p*-Value
	Low-Protein Group	High-Protein Group	
DM %	64.01 ± 0.312	64.63 ± 0.395	0.237
OM %	65.99 ± 0.354	66.61 ± 0.406	0.270
CP %	63.80 ± 0.692	66.16 ± 0.497	0.015
NDF %	62.16 ± 0.609	63.72 ± 0.378	0.047
ADF %	39.65 ± 1.077	40.23 ± 0.463	0.241

DM, dry matter; OM, Organic matter; CP, Crude protein; NDF, Neutral detergent fiber; ADF, Acid detergent fiber. Values are presented as the least-squares means with the standard deviation of the means.

**Table 4 animals-13-00367-t004:** Effects of the dietary protein level on the plasma biochemical indexes of growing yaks.

Item	Treatment		*p*-Value
	Low-Protein Group	High-Protein Group	
TP g/L	59.59 ± 0.767	60.40 ± 1.329	0.610
ALB g/L	28.56 ± 0.598	28.70 ± 0.518	0.801
GLOB g/L	31.04 ± 0.811	31.70 ± 0.715	0.551
GLU mmol/L	4.34 ± 0.110	4.97 ± 0.145	0.004
TG nmol/L	0.098 ± 0.004	0.102 ± 0.005	0.568
BUN mmol/L	2.96 ± 0.164	3.42 ± 0.0.104	0.038
ALT U/L	3.62 ± 0.101	2.89 ± 0.078	<0.001
AST U/L	44.73 ± 1.217	39.52 ± 0.659	0.002
ALP U/L	41.75 ± 0.456	39.26 ± 0.674	0.148

TP, total protein; ALB, Albumin; GLOB, Globulin; GLU, glucose; TG, total glyceride; BUN, urea nitrogen; ALT, Alanine aminotransferase; AST, Aspartate aminotransferase; ALP, Alkaline phosphatase. Values are presented as the least-squares means with the standard deviation of the means.

**Table 5 animals-13-00367-t005:** Effects of the dietary protein level on the plasma antioxidant indexes of growing yaks.

Item	Treatment		*p*-Value
	Low-Protein Group	High-Protein Group	
T-AOC U/mL	10.36 ± 0.365	10.96 ± 0.352	0.266
SOD U/mL	118.24 ± 3.627	138.64 ± 4.764	0.004
GSH-Px U/mL	182.45 ± 10.113	204.32 ± 10.496	0.156
MDA mmol/L	5.37 ± 0.108	4.66 ± 0.141	0.001

T-AOC, total antioxidant capacity; SOD, superoxide dismutase; GSH-Px, glutathione peroxidase; MDA, malondialdehyde. Values are presented as the least-squares means with the standard deviation of the means.

**Table 6 animals-13-00367-t006:** Effects of the dietary protein level on the plasma cytokine and immunoglobulin content of growing yaks.

Item	Treatment		*p*-Value
	Low-Protein Group	High-Protein Group	
IL-4 ng/L	73.78 ± 2.173	75.19 ± 1.566	0.609
IL-10 ng/L	98.43 ± 5.180	109.00 ± 3.159	0.105
IL-2 ng/L	49.35 ± 2.744	62.40 ± 3.118	0.007
IL-6 ng/L	91.60 ± 3.337	89.36 ± 2.615	0.605
IFN-γ ng/L	115.02 ± 5.499	99.27 ± 1.823	0.017
TNF-α ng/L	84.49 ± 1.529	77.85 ± 2.342	0.032
IgA g/L	0.64 ± 0.015	0.68 ± 0.026	0.299
IgG g/L	0.20 ± 0.007	0.23 ± 0.004	0.012
IgM g/L	0.07 ± 0.001	0.07 ± 0.001	0.787

IL-2, Interleukin 2; IL-4, Interleukin 4; IL-6, Interleukin 6; IL-10, Interleukin 10; IFN-γ, interferon-γ; TNF-α, tumor necrosis factor α; IgA, Immunoglobulin A; IgG, Immunoglobulin G; IgM, Immunoglobulin M. Values are presented as the least-squares means with the standard deviation of the means.

**Table 7 animals-13-00367-t007:** Effects of the dietary protein level on the rumen fermentation characteristics of growing yaks.

Item	Treatment		*p*-Value
	Low-Protein Group	High-Protein Group	
pH	6.63 ± 0.053	6.71 ± 0.062	0.084
MCP g/dL	49.53 ± 1.187	53.56 ± 1.425	0.047
NH_3_-*n* g/dL	4.60 ± 0.130	5.48 ± 0.105	<0.001
TVFA mmol/L	64.43 ± 0.902	67.88 ± 0.642	0.008
Acetate %	65.74 ± 0.527	65.31 ± 0.458	0.550
Propionate %	21.05 ± 0.519	22.10 ± 0.347	0.115
Butyrate %	9.76 ± 0.320	8.99 ± 0.267	0.088
Acetate:Propionate	3.14 ± 0.093	2.97 ± 0.058	0.127

MCP, microbial crude protein; NH_3_-*n*, ammonium nitrogen; TVFA, total volatile fatty acids. Values are presented as the least-squares means with the standard deviation of the means.

**Table 8 animals-13-00367-t008:** Different metabolites content in the plasma of yaks fed low- and high-protein diets.

Compounds	MW	RT	VIP	FC	*p* Value
Guanosine	283	1.707	1.446	2.145	0.007
L-Isoleucine	131	1.327	1.34	2.143	0.008
Acetone	58	0.974	1.354	2.135	0.002
Piperidine	85	1.009	1.476	2.125	0.004
1-Benzazole	117	5.908	1.356	1.982	0.01
Beta-Alanine	89	0.925	1.547	1.893	0.002
D-Serine	105	0.9	1.354	1.874	0.002
Harmaline	214	6.551	1.462	1.822	0.000
L-Tyrosine	181	1.188	1.469	1.786	0.002
Epinephrine	183	4.913	1.44	1.764	0.006
Xanthine	152	2.054	1.005	1.715	0.000
Adenosine	267	1.421	1.008	1.706	0.001
Indole-3-ethanol	161	6.061	1.021	1.698	0.001
Spermidine	145	0.907	1.786	1.684	0.008
Spermine	202	0.765	1.764	1.684	0.010
Histamine	111	0.957	1.457	1.675	0.006
Cytidine	243	0.927	1.576	1.657	0.008
Nandrolone phenpropionate	406	6.943	1.301	1.652	0.000
Guanine	151	1.711	1.785	1.567	0.009
Kynurenic acid	189	5.365	1.56	1.564	0.004
L-Tryptophan	204	4.197	1.167	1.556	0.009
gamma-Aminobutyric Acid	103	0.922	1.453	1.476	0.008
Thymine	126	1.725	1.678	1.457	0.003
Valine	117	0.956	1.765	1.456	0.008
L-Glutamic acid	147	0.942	1.245	1.367	0.008
(R)(-)-Allantoin	158	0.977	1.487	1.346	0.009
Creatinine	113	0.853	1.547	1.335	0.001
Cyclopentanone	84	4.097	1.357	0.482	0.007
Triethanolamine	149	20.11	1.049	0.459	0.009
Adenine	135	1.531	1.085	0.454	0.001
2’-Deoxyadenosine	251	1.526	1.071	0.452	0.001
*n*-Acetylserotonin	218	5.162	1.476	0.432	0.002
L-Phenylalanine	165	2.546	1.056	0.431	0.011
Benzamide	121	4.879	1.516	0.43	0.001
Hypoxanthine	136	1.724	1.521	0.417	0.022
3-Methylcrotonylglycine	157	1.096	1.576	0.407	0.001
(R)-Acetoin	88	4.834	1.362	0.405	0.000
3-Methyldioxyindole	163	5.225	1.069	0.369	0.001
(R)-3-((R)-3-Hydroxybutanoyloxy) butanoate	190	4.734	1.157	0.363	0.000
Stearamide	283	15.914	1.37	0.349	0.033
Norepinephrine	169	4.4	1.763	0.342	0.001
L-Methionine	149	1.111	1.334	0.335	0.003
Uracil	112	1.181	1.557	0.334	0.004
Saccharin	183	4.81	1.299	0.324	0.000
Indole-3-acetic acid	175	6.027	1.009	0.303	0.001
*n*-Undecanoylglycine	243	13.97	1.415	0.21	0.034
Sulcatone	126	2.575	1.64	0.163	0.020
Dicloralurea	352	0.283	2.207	0.06	0.009

All different metabolites listed here are those VIP > 1, fold change >2 or <0.5 and *p*-value < 0.05. MW, molecular weight; RT, retention time. VIP, Variable importance for the projection; FC, fold change.

**Table 9 animals-13-00367-t009:** Metabolic pathways and significant different metabolites that were enriched in the pathways of growing yaks fed a low- and high-protein diet.

Pathway Name	Metabolites
Protein digestion and absorption (11)	Valine,Piperidine,L-Tyrosine,L-Tryptophan,L-Phenylalanine,L-Methionine,L-Isoleucine,L-Glutamic acid,1-Benzazole,Histamine,Beta-Alanine
Arginine and proline metabolism (8)	Spermine, Spermidine, *n*-Carbamoylputrescine, L-Glutamic acid, gamma-Glutamyl-gamma-aminobutyraldehyde, gamma-Aminobutyric acid, D-Proline, Creatinine
Tryptophan metabolism (8)	Xanthurenic acid, *n*-Acetylserotonin, L-Tryptophan, Kynurenic acid,1-Benzazole,Indole-3-ethanol,Indole-3-acetic acid,3-Methyldioxyindole
Purine metabolism (8)	Xanthine,Hypoxanthine,Guanosine,Guanine,Adenosine,Adenine,2’-Deoxyadenosine,(R)(-)-Allantoin
Butanoate metabolism (5)	L-Glutamic acid, gamma-Aminobutyric Acid,Acetone,(R)-Acetoin,(R)-3-((R)-3-Hydroxybutanoyloxy)butanoate
Taste transduction (5)	Saccharin, Norepinephrine, L-Glutamic acid, gamma-Aminobutyric acid, D-Serine
Pyrimidine metabolism (4)	Uracil, Thymine, Cytidine, Beta-Alanine
Pantothenate and CoA biosynthesis (4)	Valine, Uracil, Spermine, Beta-Alanine
Glutathione metabolism (3)	Spermine, Spermidine, L-Glutamic acid
Renin secretion (3)	Norepinephrine, Epinephrine, Adenosine

The number in parentheses indicates the number of metabolites identified in the pathway.

## Data Availability

The data presented in this study are available in the article.

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
