# Peer review of "Effects of Different Dietary Protein Level on Growth Performance, Rumen Fermentation Characteristics and Plasma Metabolomics Profile of Growing Yak in the Cold Season"

_animals, 2023, doi:10.3390/ani13030367_

Round 1
Reviewer 1 Report
Although this study can provide relevant data to our field and the authors present an interesting study investigating the effects of different levels of protein dietary on growth and rumen fermentation and metabolic profiles of Yak under cold condition, there are several issues which compromise the value of the manuscript and need to address these issues before the manuscript is publishable.
1- This study is a 2×2 factorial design (protein level and gender), however, the authors did not indicate the effects of gender (male and female) on obtained results in this study!!
2- The authors never evaluated animal measures of cold condition, so do not know in fact if the animals were under cold stress or not. I suggest that the authors need to provide some meteorological data.
3- What was main reason for using these 2 levels of protein, especially 9% of CP because I think this level of protein is not proper for growing animals, it is too low!!
4- The feces and urine were collected and measured? And how the authors separated the urine from feces for female animals? Please give more details about the used methodology!!
Author Response
Although this study can provide relevant data to our field and the authors present an interesting study investigating the effects of different levels of protein dietary on growth and rumen fermentation and metabolic profiles of Yak under cold condition, there are several issues which compromise the value of the manuscript and need to address these issues before the manuscript is publishable.
Q: This study is a 2×2 factorial design (protein level and gender), however, the authors did not indicate the effects of gender (male and female) on obtained results in this study!!
A: I think the reviewer have a little misunderstanding, this trial is not a 2×2 two-factor trial design. We know that ADG of male are higher than female. Although we used both male and female, this was only to imitate the group effect under actual condition, not to examine the differences between male and female.
Q: The authors never evaluated animal measures of cold condition, so do not know in fact if the animals were under cold stress or not. I suggest that the authors need to provide some meteorological data.
A: Yaks live on the Qinghai-Tibetan Plateau, where the animal experiments was conducted, and snowfall begins every October, and if the yaks are grazing, they have no pasture to eat at all. At night, the temperature is low, and can reach minus 20 ℃, however during the day when the sun comes out, the temperature is high, sometimes it can reach 25 ℃. The temperature difference there between day and night is large. We gave a general description of the geographical location and climatic conditions in 2.2 section Study Location in L100-103. Although there are no specific indicators of animals in this article to indicate that animals were in a state of cold stress, it is a common sense that yaks in this period are under cold stress. Moreover, in the Qinghai-Tibetan Plateau, there are actually only two seasons, the cold season (November to April of the next year) and the warm season (May to October). We added the cold season to the article title, mainly wanted to distinguish it from the warm season.
Q: What was main reason for using these 2 levels of protein, especially 9% of CP because I think this level of protein is not proper for growing animals, it is too low!!
A: We actually used 10% CP in the low protein group. Because yaks generally grazing on pasture with CP less than 10%, but in this case, yaks can still grow healthily, therefore we chose 10% CP. It is true that 10% CP is too low for other growing animals, but it is not necessarily low for yaks.
Q: The feces and urine were collected and measured? And how the authors separated the urine from feces for female animals? Please give more details about the used methodology!!
A: Only part of feces was collected, we just conducted nutrients digestibility trial, not metabolism trial. We added the detailed description of feces collection in L120-122
Reviewer 2 Report
This study looked at how two levels of dietary protein affected animal performance and blood metabolites in growing Yaks during the cold season.
Because the feed conversion ratio (FCR) did not differ between the two groups, the authors must clearly explain it. Although the low protein group's feed intake and average daily gains were lower than those in the high protein group, this does not imply that the high protein level is superior.
As far as we know, increasing dietary protein improves feed intake and rumen fermentation; however, if FCR does not differ, how can you conclude that feeding high protein levels (14%) for growing Yak in cold season is beneficial?
You should include the amount of protein and energy consumed by both groups in Table 2 and then discuss their needs for maintenance and growth. Energy and protein utilization efficiency will differ between the two groups.
Although the majority of metabolites in blood appeared to be better in the high protein group than in the low protein group, those values were within normal ranges. Increasing dietary protein for growing Yak raised indoors during the cold season may not be advantageous.
Author Response
This study looked at how two levels of dietary protein affected animal performance and blood metabolites in growing Yaks during the cold season.
Q: Because the feed conversion ratio (FCR) did not differ between the two groups, the authors must clearly explain it. Although the low protein group's feed intake and average daily gains were lower than those in the high protein group, this does not imply that the high protein level is superior. As far as we know, increasing dietary protein improves feed intake and rumen fermentation; however, if FCR does not differ, how can you conclude that feeding high protein levels (14%) for growing Yak in cold season is beneficial? You should include the amount of protein and energy consumed by both groups in Table 2 and then discuss their needs for maintenance and growth. Energy and protein utilization efficiency will differ between the two groups.
A: I'm really sorry, our FCR calculation was wrong, and the other 2 reviewers also pointed out this problem. The FCR should be 12.30 and 10.65 respectively, and there was a statistically significant difference. Therefore, the latter problem should not exist. We revised the result description, pls see L204-205.
Q: Although the majority of metabolites in blood appeared to be better in the high protein group than in the low protein group, those values were within normal ranges. Increasing dietary protein for growing Yak raised indoors during the cold season may not be advantageous.
A: We conducted this trial for yaks raised indoor barn, and the blood biochemistry indexes of both groups was within the normal range, which also indicated that the two protein levels we used were also within the acceptable range for yaks. We found that the high-protein group increased ADFI and ADG and improved FCR, so increasing dietary protein levels was beneficial for yak raised indoor in cold season.
Reviewer 3 Report
This manuscript described a nutrition experiment evaluating the protein requirements of yak. Overall, the experiment was well performed with some interesting results. The English grammar does need some improvement.
No major flaws were detected.
Specific comments:
L16 - this sentence sounds like there is no data on yaks because they live on the Tibetan plateau. I do not think that is what you are trying to say.
L17 - I am not sure what stronger nitrogen utilization means. please choose a different word
L18 - you should be specific about what other species you are comparing yak to. I think you mean other species in genus Bos, but it is not clear
throughout manuscript - phrasing of treatment descriptions is awkward in some places. For example, use 'yaks fed the low-protein diet' instead of 'diet of low protein'
throughout manuscript - use greater and lesser instead of higher and lower when referring to differences between treatments
L35 - change to 'than that of yaks fed the low protein diet'
L99 - is this range in temperature the range in the daily average temperature or the lowest and highest temperature observed across all days
L107 - change to 'were assigned to individual pens'
L122 - change to ' were weighed'
L129 - did the fecal collection in the last 4 days alter feed intake patterns that could have affected the final body weight on day 56
L130-132 - this description sounds like total feces were collected rather than a single hand sample, but that is not clearly stated
L135 - if individual intake was measured and total feces collected, why was acid detergent insoluble ash measured?
L143 - here you say plasma was analyzed, but in the rest of the paper you refer to serum
L184 - what time of day were rumen samples taken and when relative to feeding? How many times were rumen samples taken?
L185 - about 75% of rumen microorganisms are attached to feed particles so just filtering out the fluid may have biased your measurement of microbial crude protein
L205-216 - this is a good description of the metabolomic data analysis, but there is no description of statistical analysis of other data (growth, intake, plasma, rumen). please add
Table 2 - are these standard errors or standard deviations reported along with the LSmeans. please indicate in the table title or footnote for all tables
Table 2 - I calculate a FCR for the high protein group of 10.65 instead of 12.11. please double check this value
L277 - why both PCA and OPLS-DA? what is the purpose of this analysis because you do not go on to discuss it. the discussion focuses on the KEGG pathways involved
L282 - please define abbreviations R2X, R2Y, Q2
L283 - change to 'model is valid'
Table 8 - what is the order of the compounds? it would be easier for the reader if the compounds were ordered by FC largest to smallest
Figure 2 - I am not sure what this figure is telling me. There is no treatment designation and I assume large circles indicate more differentially abundant compounds were identified in that pathway, but it is not clear. It is also not clear what the color (yellow to orange) indicates. There is little explanation of these data.
Figure 2 caption - change to 'Metabolome map view' or 'Metabolome map'
L317 - did you measure maintenance energy requirements of the yaks or are you basing this statement on previous research. if so please provide a reference
L320 - a lesser proportion of the daily energy intake would be used for maintenance
L324-325 - I suspect that the increased ADFI is due to the increase in NDF digestibility, but maybe the taste transduction pathway plays a role. Please provide a reference indicating the role of the taste transduction pathway
L337 - change to 'yak from the high-protein'
L338 - I am not sure what metabolize more vigorously means. please choose different language
L350 - what is the 10% group? I assume you mean the low protein group, but please use the same name for each treatment group throughout
L368 - instead of saying there was 'no doubt' please say that it was 'expected' that the protein pathway would be affected
L371 - please do not use etc. please list out all of the compounds. do this here and the other 2 instances of etc.
L414-421 - This paragraph is restating the individual results. Please provide a broader conclusion
Author Response
This manuscript described a nutrition experiment evaluating the protein requirements of yak. Overall, the experiment was well performed with some interesting results. The English grammar does need some improvement.
No major flaws were detected.
Specific comments:
L16 - this sentence sounds like there is no data on yaks because they live on the Tibetan plateau. I do not think that is what you are trying to say.
Changed. Pls see L20.
L17 - I am not sure what stronger nitrogen utilization means. please choose a different word
Changed. Pls see L21.
L18 - you should be specific about what other species you are comparing yak to. I think you mean other species in genus Bos, but it is not clear
Changed. Pls see L22.
throughout manuscript - phrasing of treatment descriptions is awkward in some places. For example, use 'yaks fed the low-protein diet' instead of 'diet of low protein'
throughout manuscript - use greater and lesser instead of higher and lower when referring to differences between treatments
Changed according to suggestion throughout manuscript.
L35 - change to 'than that of yaks fed the low protein diet'
Changed. Pls see L40.
L99 - is this range in temperature the range in the daily average temperature or the lowest and highest temperature observed across all days
It is the lowest and highest temperature observed during the experimental period.
L107 - change to 'were assigned to individual pens'
Changed. Pls see L109.
L122 - change to ' were weighed'
Changed. Pls see L113.
L129 - did the fecal collection in the last 4 days alter feed intake patterns that could have affected the final body weight on day 56.
In order to reduce pollution, before we carried out the digestion test, a thick plastic film was spread on the floor of the pen, and the feces were collected into the plastic bucket with a small shovel immediately after the yak defecation. This was how we conducted the digestibility trial. We dare not say that there is no impact on growth performance at all, but this should be in the normal range that large animal digestibility trial can bear.
L130-132 - this description sounds like total feces were collected rather than a single hand sample, but that is not clearly stated
Changed. Pls see L120-122.
L135 - if individual intake was measured and total feces collected, why was acid detergent insoluble ash measured ?
We did not collect all the feces, but a part of feces, approximately 3-5% of the total feces, and we calculated the nutrients digestibility using Ash as an internal standard.
Changed. Pls see L 122-123.
L143 - here you say plasma was analyzed, but in the rest of the paper you refer to serum
Sorry, we are not rigorous. We thought plasma and serum were synonymous, and now we know that they mean differently. We changed it all to plasma.
L184 - what time of day were rumen samples taken and when relative to feeding? How many times were rumen samples taken?
On d 56, approximately 150 mL rumen fluid was collected using an oral stomach tube at 2, 4, 6 h post morning feeding. Pls see L169-170.
L185 - about 75% of rumen microorganisms are attached to feed particles so just filtering out the fluid may have biased your measurement of microbial crude protein
We agree with the reviewer. Actually, rumen microorganisms consist of 3 parts, liquid phase, solid phase, and rumen wall. The microorganisms measured in the samples obtained with the filter cloth represent only liquid-phase microorganisms, but it has long been accepted by the academic community for many years.
L205-216 - this is a good description of the metabolomic data analysis, but there is no description of statistical analysis of other data (growth, intake, plasma, rumen). please add
Added. Pls see L188-189.
Table 2 - are these standard errors or standard deviations reported along with the LS means. please indicate in the table title or footnote for all tables.
Added the footnote for the Tables.
Table 2 - I calculate a FCR for the high protein group of 10.65 instead of 12.11. please double check this value
We are awfully sorry, the FCR data was presented incorrectly. The other 2 reviewers also found this issue. It's been changed now L204-205.
L277 - why both PCA and OPLS-DA? what is the purpose of this analysis because you do not go on to discuss it. the discussion focuses on the KEGG pathways involved
PCA is to identify whether the two groups can be distinguished, OPLS-DA also has a similar function with PCA, but OPLS-DA also did VIP analysis at the same time, that is, to identify the metabolites with greater impact, and these metabolites with greater influence almost are the main metabolic intermediates of KEGG. The PCA and OPLS-DA results are visual, there really isn't much to discuss in themselves.
L282 - please define abbreviations R2X, R2Y, Q2
Added. Pls see L241-243. R2X and R2Y represents the explanatory rate of the model in the X and Y-axis direction. Q2 represents the predictive rate of the model.
L283 - change to 'model is valid'
Changed. Pls see L243.
Table 8 - what is the order of the compounds? it would be easier for the reader if the compounds were ordered by FC largest to smallest.
Changed. Pls see Table 8.
Figure 2 - I am not sure what this figure is telling me. There is no treatment designation and I assume large circles indicate more differentially abundant compounds were identified in that pathway, but it is not clear. It is also not clear what the color (yellow to orange) indicates. There is little explanation of these data. Figure 2 caption - change to 'Metabolome map view' or 'Metabolome map'
Changed. X-axis represents pathway impact and Y-axis represents p value. The larger size of circle indicates more metabolites enriched in that pathway and the larger abscissa indicates higher pathway impact values. The darker color indicates the smaller p values. Pls see L 699-702.
L317 - did you measure maintenance energy requirements of the yaks or are you basing this statement on previous research. if so please provide a reference
We thought about it carefully and felt that L317 should be deleted, pls see L256. Because it was said later that the isothermal zone is 8-14 ℃, and since the average temperature during the animal experiment is 0.6, then the maintenance requirement should definitely increase, and this was addressed later.
L320 - a lesser proportion of the daily energy intake would be used for maintenance
Changed. Pls see L259.
L324-325 - I suspect that the increased ADFI is due to the increase in NDF digestibility, but maybe the taste transduction pathway plays a role. Please provide a reference indicating the role of the taste transduction pathway.
We agree with the reviewer that there should be a literature on increasing dietary protein level to activate animal’s taste transduction pathway, however we did not find proper literature. This may be the highlight of this article.
L337 - change to 'yak from the high-protein'
Changed. Pls see L273.
L338 - I am not sure what metabolize more vigorously means. please choose different language
Changed to more efficiently. Pls see L275.
L350 - what is the 10% group? I assume you mean the low protein group, but please use the same name for each treatment group throughout
Changed. pls see L284.
L368 - instead of saying there was 'no doubt' please say that it was 'expected' that the protein pathway would be affected
Changed pls see L301.
L371 - please do not use etc. please list out all of the compounds. do this here and the other 2 instances of etc.
Added in 3 places. Pls see L304, 307, 331.
L414-421 - This paragraph is restating the individual results. Please provide a broader conclusion
Changed. Pls see L342-346.
Reviewer 4 Report
KEY OBSERVATIONS: 1. The amount of phosphorus is observed to be very high for ruminants, therefore, in later experiments they should justify why they use such high levels and without a 2 to 1 relationship for ruminants. 2. The feed conversion ratio must be 5.51 divided by 0.44 and this is equal to 12.30, and in the manuscript it is observed 12.12. Therefore, it is recommended to correct these numbers. 3. In the same table, the feed conversion ratio should be 6.60 divided by 0.62 and this is equal to 10.65 and the manuscript shows 12.11, which I consider to be incorrect. Thoroughly review these calculations. 4. The work is interesting and I think it can be publishable if corrections or clarifications are considered.
Author Response
KEY OBSERVATIONS: 1. The amount of phosphorus is observed to be very high for ruminants, therefore, in later experiments they should justify why they use such high levels and without a 2 to 1 relationship for ruminants.
A: The phosphorus level in Table 1 was wrong and has been modified. Pls see Table 1.
- The feed conversion ratio must be 5.51 divided by 0.44 and this is equal to 12.30, and in the manuscript it is observed 12.12. Therefore, it is recommended to correct these numbers. 3. In the same table, the feed conversion ratio should be 6.60 divided by 0.62 and this is equal to 10.65 and the manuscript shows 12.11, which I consider to be incorrect. Thoroughly review these calculations.
A: Thank you very much for pointing us out this error, and we are ashamed that this most basic data was calculated incorrectly. Also, 2 other reviewers also discovered the bug at the same time. One reviewer even said that the FCR has not decreased, so the conclusion cannot be said that increasing protein levels is beneficial for yak farming in cold season. However, we have now corrected it, and we can now draw conclusions like this.
- The work is interesting and I think it can be publishable if corrections or clarifications are considered.
A: Yak trials are usually conducted on the plateau, it is very difficult to carry out the experiment, and the researchers worked very hard. Thanks a lot to the reviewers for your recognition of our work, and we hope to show more yak research results to people in this industry.
Round 2
Reviewer 1 Report
the Authors have modified all comments made by reviewers, but the manuscript is needs to follow the journal formats and instructions